# Tracking $\mathcal{R}$ of COVID-19: A new real-time estimation using the Kalman filter

**Francisco Arroyo-Marioli[1], Francisco Bullano[1], Simas Kucinskas [ID][2]\*, Carlos Rondón-Moreno [ID][1]**

1 Central Bank of Chile, Santiago, Chile, 2 Humboldt University of Berlin, Berlin, Germany

\* simas.kucinskas@hu-berlin.de

**Data Availability Statement:** All relevant data are within the manuscript and its Supporting information files.

**Funding:** The author(s) received no specific funding for this work.

## Abstract

We develop a new method for estimating the effective reproduction number of an infectious disease ($\mathcal{R}$) and apply it to track the dynamics of COVID-19. The method is based on the fact that in the SIR model, $\mathcal{R}$ is linearly related to the growth rate of the number of infected individuals. This time-varying growth rate is estimated using the Kalman filter from data on new cases. The method is easy to implement in standard statistical software, and it performs well even when the number of infected individuals is imperfectly measured, or the infection does not follow the SIR model. Our estimates of $\mathcal{R}$ for COVID-19 for 124 countries across the world are provided in an interactive online dashboard, and they are used to assess the effectiveness of non-pharmaceutical interventions in a sample of 14 European countries.

## Introduction

The effective reproduction number ($\mathcal{R}$) plays a central role in the epidemiology of infectious diseases. $\mathcal{R}$ is defined as the average number of secondary cases produced by a primary case [1–3]. The effective reproduction number varies over time, due to the depletion of susceptible individuals as well as changes in other factors, including control measures, contact rates, and climatic conditions. The *basic reproduction number*, denoted by $\mathcal{R}_0$, measures the average number of secondary cases produced by a primary case when the population is fully susceptible [4, 5]. Analogously to the effective reproduction number, the basic reproduction number is also affected by multiple variables [6].

In standard models, the number of infected individuals increases as long as $\mathcal{R} > 1$. Real-time estimates of $\mathcal{R}$ are therefore essential for public policy decisions during a pandemic [7, 8]. Such estimates can be used to study the effectiveness of non-pharmaceutical interventions (NPIs), or assess what fraction of the population needs to be vaccinated to reach herd immunity [9–11]. Some social scientists have argued that $\mathcal{R} < 1$ should be viewed as a fundamental constraint on public policy during the current COVID-19 pandemic [12].

In this paper we develop a new method to estimate $\mathcal{R}$ in real time. The method exploits the fact that in the benchmark SIR model, $\mathcal{R}$ is linearly related to the growth rate of the number of infected individuals [13]. Our estimation procedure consists of three steps. First, we use data on new cases to construct a time series of how many individuals are infected at a given point

**Competing interests:** The authors have declared that no competing interests exist.

in time. Then, we estimate the growth rate of this time series with the Kalman filter. In the final step, we leverage the theoretical relationship given by the SIR model to obtain $\mathcal{R}$ from the estimated growth rate. We show theoretically that the estimates are not sensitive to potential model misspecification, and they are fairly accurate even when new cases are imperfectly measured.

We apply our methodology to estimate the $\mathcal{R}$ of COVID-19 in real-time. Our estimates for 124 countries across the world are provided in an online dashboard and can be explored interactively [14]. In empirical applications, we use these estimates to calculate the basic reproduction number ($\mathcal{R}_0$) and evaluate the effects of NPIs in reducing $\mathcal{R}$ for a sample of 14 European countries.

Under our baseline assumption that the serial interval for COVID-19 is seven days, we estimate the basic reproduction number ($\mathcal{R}_0$) to be 2.66 (95% CI: 1.98–3.38). Next, we find that lockdowns, measures of self-isolation, and social distancing all have a statistically significant effect on reducing $\mathcal{R}$. However, we also demonstrate the importance of accounting for voluntary changes in behavior. In particular, we document that most of the decline in mobility in our sample happened before the introduction of lockdowns. Failing to account for voluntary changes in behavior leads to substantially over-estimated effects of NPIs.

## Related literature

There are two broad classes of methods that can be used to estimate $\mathcal{R}$ in real time [2, 5, 15]. First, one can estimate a fully-specified epidemiological model and then construct a model-implied time series for $\mathcal{R}$ [10, 16–18]. Second, one may use approaches that leverage information on the serial interval of a disease (i.e., time between onset of symptoms in a case and onset of symptoms in his/her secondary cases) [1, 3, 19]. For example, imagine a disease with a fixed serial interval of, say, three days. In that case, we could estimate $\mathcal{R}$ by simply dividing the number of new cases today by the number of new cases three days ago. Cori et al [3] exploit this idea to develop a Bayesian estimator that accounts for the randomness in the onset of infections as well as variation in the serial interval. This method is implemented in a popular R package `EpiEstim` [20].

The method proposed in this paper attempts to strike a balance between the two approaches mentioned above. Although our estimator is derived from standard epidemiological theory, we use the smallest amount of theoretical structure that is necessary to obtain our estimator. In particular, the theoretical relationship used to derive our estimator is exactly valid not only in the standard SIR model with constant parameters, but also in the SIS model and a generalized SIR model with time-varying parameters and stochastic shocks. Relative to the existing literature, our estimator does not need any statistical tuning parameters, and it does not require parametric assumptions on the distribution of new cases (such as assuming that new cases are Poisson distributed). For example, the method of Cori et al [3] assumes that $\mathcal{R}$ is constant over fixed windows of duration $\tau$; $\tau$ effectively becomes a tuning parameter that needs to be chosen by the user. Our approach and its mathematical derivation share some similarities with the estimator proposed by Bettencourt and Ribeiro [21].

A key advantage of using the Kalman filter for estimating $\mathcal{R}$ is that valid confidence bounds are readily obtained. Explicitly accounting for the dynamics in $\mathcal{R}$ via the state equation ensures that the estimated effective reproduction numbers are not excessively volatile, with the optimal amount of filtering estimated from the data. In addition, the Kalman smoother allows the researcher to use full-sample information efficiently when estimating $\mathcal{R}$. Finally, our method can be used with both classical and Bayesian techniques, as we demonstrate in the empirical application.

## Materials and methods

### Data sources

We use data on COVID-19 cases from the John Hopkins CSSE repository [22]. For some of our statistical analyses, we also use data on the number of daily tests per capita collected by *Our World in Data* [23], mobility data from Google's "COVID-19 Community Mobility Reports" [24], and data on NPIs collected by Flaxman et al [25, 26]. All of these datasets are publicly available online. The computer code and data used in the study are provided in S1 File.

### New real-time estimator

We now derive our estimator for the SIR model [13]. In S1 Appendix (Sections A.1 and A.2), we show that we can obtain the same estimator from an SIS model, and an SIR model with stochastic shocks.

The standard SIR model in discrete time describes the evolution of susceptible ($S_t$), infected ($I_t$), and recovered ($R_t$) individuals by the following equations [27, 28]:

$$
\begin{aligned}
S_t &= S_{t-1} - \beta_t I_{t-1} \frac{S_{t-1}}{N} \\
I_t &= I_{t-1} + \beta_t I_{t-1} \frac{S_{t-1}}{N} - \gamma I_{t-1} \\
R_t &= R_{t-1} + \gamma I_{t-1}
\end{aligned}
\tag{1}
$$

The model is stated at a daily frequency. Here, $N \equiv S_t + I_t + R_t$ is the population size, $\beta_t$ is the daily transmission rate, and $\gamma$ is the daily transition rate from infected to recovered. The recovered group consists of individuals who have either died or fully recovered. We allow the transmission rate $\beta_t$ to vary over time. For example, individuals may choose to to reduce their social interactions voluntarily, or they could be subject to government policy restrictions.

The *basic reproduction number*, $\mathcal{R}_0^{(t)}$, is defined as $\mathcal{R}_0^{(t)} \equiv \beta_t / \gamma$, and it gives the average number of individuals infected by a single infected individual when everyone else is susceptible. Since the transmission rate $\beta_t$ varies over time, the basic reproduction number is generally time varying as well. The *effective reproduction number*, $\mathcal{R}_t$, is defined as $\mathcal{R}_t = \mathcal{R}_0^{(t)} \times (S_{t-1}/N)$, and it equals the average number of individuals infected by a single infected individual when a fraction ($S_{t-1}/N$) of individuals is susceptible.

From Eq (1) the daily growth rate in the number of infected individuals is

$$
\mathrm{gr}(I_t) \equiv \frac{I_t - I_{t-1}}{I_{t-1}} = \gamma(\mathcal{R}_t - 1).
\tag{2}
$$

Denoting the estimated growth rate of infected individuals by $\hat{\mathrm{gr}}(I_t)$, and given a value for the transition rate $\gamma$, the plug-in estimator for the effective reproduction number is

$$
\hat{\mathcal{R}}_t = 1 + \frac{1}{\gamma} \hat{\mathrm{gr}}(I_t).
\tag{3}
$$

For the estimator to be feasible, we need to (i) calibrate the transition rate from infectious to recovered, $\gamma$; and (ii) estimate the growth rate of $I_t$. There are two potential strategies for choosing $\gamma$. First, we can use external medical evidence given that $\gamma^{-1}$ is the average infectious period. Second, information on the serial interval of the disease can be employed, given that the serial interval in the SIR model also equals $\gamma^{-1}$ [29].

To estimate the growth rate of $I_t$ empirically, we first construct a time series for $I_t$ from data on new cases. The SIR model in Eq (1) implies that

$$I_t = (1 - \gamma)I_{t-1} + \text{new cases}_t. \tag{4}$$

We initialize $I_t$ by $I_0 = C_0$ where $C_0$ is the total number of infectious cases at some initial date, and then construct subsequent values of $I_t$ recursively.

Given the time series for $I_t$, we use standard Kalman-filtering tools to smooth the observed growth rate of $I_t$. In particular, we specify the following state-space model for the growth rate of $I_t$:

$$\begin{aligned} \text{gr}(I_t) &= \gamma(\mathcal{R}_t - 1) + \varepsilon_t, \quad \varepsilon_t \sim \text{i.i.d. } \mathcal{N}(0, \sigma_\varepsilon^2) \\ \mathcal{R}_t &= \mathcal{R}_{t-1} + \eta_t, \qquad \eta_t \sim \text{i.i.d. } \mathcal{N}(0, \sigma_\eta^2) \end{aligned} \tag{5}$$

We estimate $\mathcal{R}_t$ by the Kalman smoother [30]. The Kalman smoother provides optimal estimates of $\mathcal{R}_t$ (in the sense of minimizing mean-squared error) given the full-sample information on $\text{gr}(I_t)$, provided that the data are generated by the model in Eq (5).

To estimate the unknown parameters $\sigma_\varepsilon^2$ and $\sigma_\eta^2$ in Eq (5), both classical and Bayesian methods can be used. However, sample sizes are usually limited in practice, especially early on in the epidemic. Hence, incorporating prior knowledge generally leads to better-behaved estimates. The state-space model above—also known as the local-level model—can also be thought as a model-based version of exponentially-weighted moving-average smoothing [31].

The state-space model in Eq (5) can be viewed as a reduced-form time-series specification. The local-level model can capture fairly rich dynamic patterns in the data [30, 32]. In addition, in S1 Appendix (Section A.3), we provide a theoretical rationale for the local-level specification. In particular, Eq (5) arises naturally in the SIR model (in the early stages of an epidemic) when the transmission rate $\beta_t$ follows a random walk.

From Eq (4), the growth rate $\text{gr}(I_t)$ is bounded below by $(-\gamma)$. Hence, for any estimator of $\text{gr}(I_t)$ that is some weighted average of the observed growth rates, the point estimate of $\mathcal{R}_t$ is automatically non-negative. To ensure that lower confidence bounds are positive as well, we estimate the $q$-th quantile of $\mathcal{R}_t$ by $\max\{0, 1 + \gamma^{-1}\hat{g}_q\}$, where $\hat{g}_q$ is an estimate of the $q$-th quantile of $\text{gr}(I_t)$. In addition (see Section A.4 in S1 Appendix), our empirical estimates remain similar when we use a modified version of the Carter-Kohn [33] algorithm which discards random draws violating the non-negativity constraint. Alternatively, it is possible to avoid this type of truncation by using non-linear filtering methods [34].

## Sensitivity to model misspecification and data problems

Tracking the evolution of $\mathcal{R}_t$ is notoriously difficult. Human-contact dynamics, testing, and changes in case definitions affect the flow and quality of the available information. In this section, we test the sensitivity of our estimator to two notable issues: (i) model misspecification; and (ii) data problems (such as reporting delays or imperfect detection of infectious individuals).

For the first issue, model misspecification, a natural concern is whether the true dynamics of the disease are well captured by the benchmark SIR model. We address this issue in two ways. First, we show that our estimator remains exactly valid in the SIS model in which individuals do not obtain immunity, and a generalized SIR model with stochastic shocks (see Sections A.1 and A.2 in S1 Appendix). In addition, provided that the average duration of infectiousness is correctly specified, we find that our estimator yields accurate results even when the true model is SEIR rather than SIR (see Section A.5 in S1 Appendix). Second, we note that the

error term $\varepsilon_t$ in the state-space model described by Eq (5) can be interpreted as model error. Therefore, our estimates as well as their confidence intervals explicitly account for (some amount of) potential misspecification.

The second issue relates to data reliability. For COVID-19, testing constraints and high asymptomatic prevalence [35–37], in particular, make it challenging to identify all infectious individuals. The simplicity of our estimator allows us to analytically characterize the effects of potential measurement error (see Section A.6 in S1 Appendix). Furthermore, we use these results to investigate the quantitative performance of the estimator in a number of empirically relevant underdetection scenarios using Monte Carlo simulations. Overall, we conclude that our method provides accurate estimates in all cases that we analyze.

## Results

In our estimations, we include all countries for which we have at least 20 daily observations after the cumulative number of confirmed COVID-19 cases reaches 100. Our sample period starts on 2020-01-23 and finishes on 2020-05-06. For the baseline estimates, we assume that people are infectious for $\gamma^{-1} = 7$ days on average, consistent with recent literature [38, 39]. This assumption also accords with the evidence on the serial interval of COVID-19. For example, Flaxman et al [25] use an average serial interval of 6.5 days. Recent studies find that estimates of the serial interval for COVID-19 generally range between 4 and 9 days [40–42]. In addition, we document that $\gamma^{-1} = 7$ leads to estimates of the basic reproduction number ($\mathcal{R}_0$) that are in line with the recent estimates in the literature [43]. However, we also investigate the effects of different choices for $\gamma$ on our results. In general, by virtue of Eq (3), changing $\gamma$ tilts the estimates of $\mathcal{R}_t$ around one, with higher values of the serial interval pushing the estimates away from one and lower values pushing the estimates towards one. For example, if $\hat{\mathcal{R}}_t = 1.5$ for $\gamma^{-1} = 7$ days, increasing the serial interval to 8 days increases the estimate to $\hat{\mathcal{R}}_t \approx 1.57$. Conversely, if $\hat{\mathcal{R}}_t = 0.5$ for $\gamma^{-1} = 7$ days, increasing the serial interval to 8 days decreases the estimate to $\hat{\mathcal{R}}_t \approx 0.43$. S1 Appendix (Section A.7) describes the details of the estimation procedure. S1 Appendix (Section A.15) also contains the GATHER checklist [44], summarizing the details of the analysis.

In S1 Appendix (Section A.8), we perform two empirical validation exercises of our estimates. First, we document that our estimates of $\mathcal{R}_t$ are predictive of future deaths. Given that deaths are arguably more accurately measured, this finding alleviates concerns regarding potential data reliability issues that could contaminate our estimates. Second, we find that past mobility data is predictive of future values of $\mathcal{R}_t$. In S1 Appendix (Section A.12), we additionally compare our estimates to those obtained using the method of Cori et al [3]. We find that our estimates are highly correlated to the estimates produced by the Cori et al method, with the average correlation coefficient across different countries equal to 0.80 (median: 0.89). Jointly, these exercises suggest that our estimates contain valuable information on the dynamics of COVID-19.

### Estimated effective reproduction numbers

Our estimates of $\mathcal{R}_t$ for selected countries are shown in Figs 1 and 2. Estimates for the remaining countries can be found in the associated dashboard [14].

Fig 1 plots the estimated effective reproduction numbers for China, Italy, and the US. In S1 Appendix (Section A.9, Fig A.7), we also provide a graph of the raw data on the growth rate of the number of infected individuals that is used for estimating $\mathcal{R}_t$. For all three countries, the estimated $\mathcal{R}_t$ is initially above 3. For China, the estimated $\mathcal{R}_t$ declined rapidly, falling below

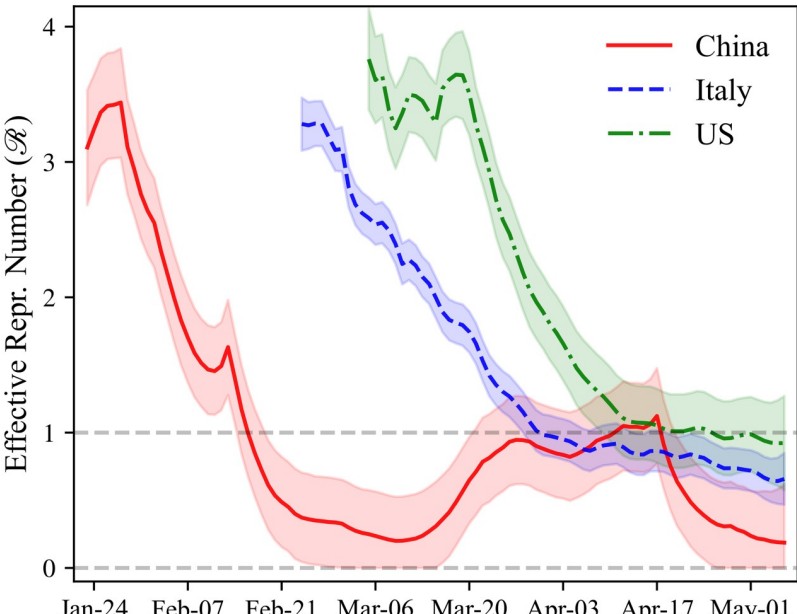

**Fig 1. $\mathcal{R}_t$ of COVID-19: China, Italy, and the US.** Estimates of the effective reproduction rate ($\mathcal{R}_t$) of COVID-19 for selected countries. The sample consists of all dates after the total number of reported cases in the country has reached 100. 65% credible bounds shown by the shaded areas.

one around the third week of February. According to our estimates, $\mathcal{R}_t$ in China fell below one 24 days after the beginning of the epidemic in the country (with the start of the epidemic defined as reaching 100 cumulative confirmed cases of COVID-19). However, the estimated $\mathcal{R}_t$ in China drifted up towards one during late March and early April, potentially caused by a

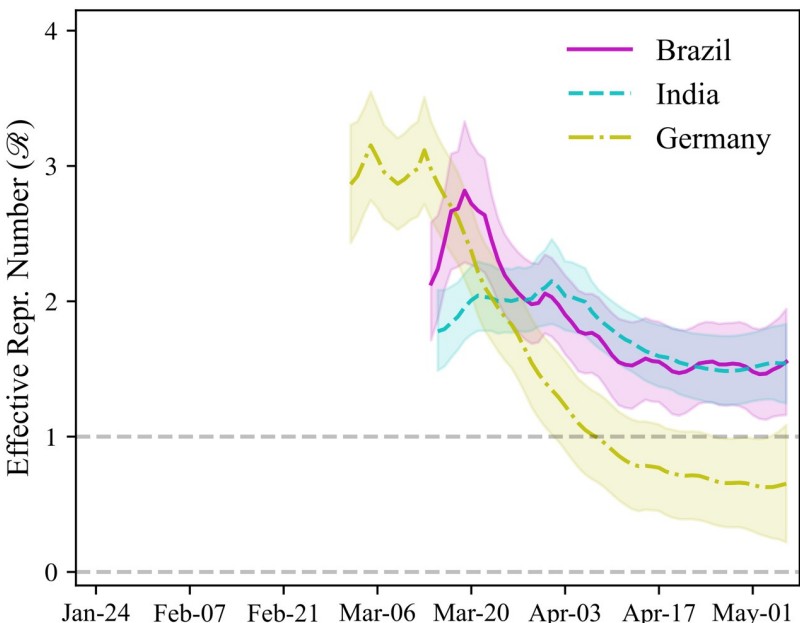

**Fig 2. $\mathcal{R}_t$ of COVID-19: Brazil, India, and Germany.** Estimates of the effective reproduction rate ($\mathcal{R}_t$) of COVID-19 for selected countries. The sample consists of all dates after the total number of reported cases in the country has reached 100. 65% credible bounds shown by the shaded areas.

wave of imported cases. Note that there is an upwards jump in the estimated $\mathcal{R}_t$ for China around the second week of February. This jump was caused by a temporary change in COVID-19 case definitions in the Hubei province in China; the new definition included clinically-diagnosed COVID-19 cases [45].

In Italy, the estimated $\mathcal{R}_t$ fell steadily since March but at a slower rate than previously observed in China, with the point estimate for Italy falling below one in early April. Our estimates indicate that it took 36 days for $\mathcal{R}_t$ to fall below one after the start of the epidemic in Italy. In the US, the point estimates of $\mathcal{R}_t$ were fairly flat in the first two weeks of the epidemic, hovering around 3.5. We note, however, that it is likely that the fraction of non-detected cases in the US went down substantially in this period, inflating the estimates of $\mathcal{R}_t$ upward. In particular, the daily number of tests conducted in the U.S went up dramatically during this period, increasing by a factor of 45 between March 8, 2020 and March 25, 2020 [23]. It took 52 days for the estimated $\mathcal{R}_t$ to fall below one for the first time in the US after the start of the epidemic, or more than twice as long as in China. The point estimate of $\mathcal{R}_t$ in the US at the end of the sample is below one and equal to 0.92 (95% CI: 0.17–1.66).

Fig 2 plots the estimated effective reproduction numbers for Brazil, India, and Germany. The pattern observed in Germany is similar to that previously seen in Italy and the United States. The estimated $\mathcal{R}_t$ in Germany falls below one 37 days after the beginning of the pandemic, almost identically to Italy. In Brazil and India, the point estimates of $\mathcal{R}_t$ are lower at the beginning of the pandemic than in the other countries plotted here. The effective reproduction numbers at the beginning of the epidemic are estimated to be 2.13 (95% CI: 0.81–3.04) in Brazil, and 1.78 (95% CI: 0.92–2.41) in India. In contrast, for example, $\mathcal{R}_t$ is estimated to be 2.86 (95% CI: 1.91–3.81) in Germany at the beginning of the pandemic. We emphasize that the estimated confidence bounds are wide, indicating substantial uncertainty about the true values of $\mathcal{R}_t$. Hence, substantial caution must be exercised when comparing the estimates of $\mathcal{R}_t$ across countries and over time.

A natural concern with any estimator of $\mathcal{R}_t$ applied to COVID-19 is that the estimator may be biased if only a fraction of all COVID-19 cases is detected. In S1 Appendix (Section A.6), we study the performance of our estimator under various assumptions on the reporting of COVID-19 cases. We show analytically that our estimator remains exactly valid even when only a fraction of all cases is detected (e.g., 10% of all cases are detected), provided that the fraction of all cases detected is constant over time. The estimates are also accurate under some other cases of misreporting. However, if the fraction of detected COVID-19 cases changes a lot over short windows of time, the estimator is biased. Finally, we investigate the performance of our estimator in a number of additional cases of imperfect reporting (such as a ramp-up in testing) that may be important in practice using Monte Carlo simulations. Overall, we conclude that our estimator is robust to potential mismeasurement of COVID-19 cases in a number of empirically-relevant scenarios.

In S1 Appendix (Section A.10, Fig A.8), we illustrate the difference between estimates of $\mathcal{R}_t$ for China obtained by the Kalman smoother—as in our baseline estimation—and the Kalman filter. Intuitively, the Kalman smoother uses information from the full sample when estimating $\mathcal{R}_t$, while the Kalman filter only uses information up to and including time $t$ [30]. While the two sets of estimates are fairly similar, the filtered estimates are substantially more volatile. In addition, the filtered estimates generally have wider credible bounds. As should be the case, the filtered and smoothed estimates are identical at the endpoint of the sample. From the perspective of epidemiological theory, the Kalman filter essentially produces what Fraser [46] refers to as the instantaneous reproduction number, while the Kalman smoother yields the case reproduction number. The estimator proposed in the present

paper therefore allows researchers to estimate the two types of reproduction numbers in a single unified framework.

In S1 Appendix (Section A.10, Fig A.8), we also demonstrate the difference between our Bayesian estimates of $\mathcal{R}_t$ and classical estimates obtained via maximum likelihood. For China, the two sets of estimates are virtually indistinguishable, indicating that the chosen priors have a small effect on the estimates. Of course, for other some countries in our sample, the data are less informative, and hence the priors have a more pronounced effect.

## Basic reproduction number

We now use our estimates of $\mathcal{R}_t$ to measure the basic reproduction number ($\mathcal{R}_0$), i.e., the average number of individuals infected by a single infectious individual when the population is fully susceptible. We estimate $\mathcal{R}_0$ by the average value of $\mathcal{R}_t$ in the first week of the epidemic.

Table 1 shows the results for a sample of 14 European countries (Austria, Belgium, Denmark, France, Germany, Greece, Italy, Netherlands, Norway, Portugal, Spain, Sweden, Switzerland, and United Kingdom), as in Flaxman et al [25]. Under our baseline assumption that the individuals are infectious for 7 days on average ($\gamma = 1/7$), we obtain an estimate of $\mathcal{R}_0 = 2.66$ (95% CI: 1.98–3.38). For COVID-19, a recent meta-study has estimated a median $\mathcal{R}_0$ of 2.79 [43], suggesting that our results are consistent with the current consensus estimates.

Table 1 also provides the estimated $\mathcal{R}_0$ under different assumptions on the duration of infectiousness (or, equivalently in the SIR model, the average serial interval). As expected, the median estimate is sensitive to the choice of $\gamma$; we find an additional day of infectiousness increases $\mathcal{R}_0$ by around 0.3.

## Assessing non-pharmaceutical interventions

Finally, we use our estimates to assess the effects of non-pharmaceutical interventions (NPIs) in the same sample of 14 European countries as in the previous section. We study a total of five NPIs: (i) lockdowns; (ii) bans of public events; (iii) school closures; (iv) mandated self-isolation when exhibiting symptoms; and (e) social distancing measures. We adopt the definitions of NPIs and their introduction dates provided by Flaxman et al [25].

We first perform an event-study exercise, inspired by event studies commonly used in economics and finance [47]. In this exercise, we compare the dynamics of the effective reproduction number before and after the introduction of a particular control measure. If the control measure is effective, we expect to observe a difference in the behavior of $\mathcal{R}_t$ after its introduction. The difference may appear as either a change in levels ("jump") or a change in trends ("kink"); the latter possibility is more likely in the present empirical context. This simple before-versus-after comparison is not free of potential bias. In particular, the comparison

**Table 1. Estimates of the basic reproduction number ($\mathcal{R}_0$).**

| Number of Days Infectious: | 5 | 6 | 7 | 8 | 9 | 10 |
|---|---|---|---|---|---|---|
| $\hat{\mathcal{R}}_0$ | 2.07 | 2.35 | 2.66 | 2.89 | 3.10 | 3.29 |
| CI Lower Bound (95%) | 1.51 | 1.75 | 1.98 | 2.17 | 2.37 | 2.54 |
| CI Upper Bound (95%) | 2.67 | 3.01 | 3.38 | 3.67 | 3.86 | 4.06 |

Estimates of the basic reproduction number ($\mathcal{R}_0$) for a sample of 14 European countries. The countries included in the sample are Austria, Belgium, Denmark, France, Germany, Greece, Italy, Netherlands, Norway, Portugal, Spain, Sweden, Switzerland, and United Kingdom. The basic reproduction number is calculated by averaging our estimates of the effective reproduction number in the first 7 days of the epidemic, where the start of the epidemic is defined as the day when the cumulative number of cases reaches 100.

implicitly assumes that the behavior of $\mathcal{R}_t$ before the intervention provides a good counterfactual for the (unobserved) future behavior of $\mathcal{R}_t$ in the absence of the intervention. Nevertheless, we find this exercise instructive as a preliminary step in our analysis.

Fig 3 plots the estimated values of $\mathcal{R}_t$ one week before and three weeks after the introduction of a lockdown. Since Sweden did not have a lockdown in the sample period considered, the figure is constructed using data from 13 countries. $\mathcal{R}_t$ declines substantially after a lockdown is introduced, going from 2.11 (95% CI: 1.84–2.38) on the day of the intervention to 0.99 (95% CI: 0.87–1.11) three weeks later. However, $\mathcal{R}_t$ is decreasing before the lockdown as well. In particular, there is no visually detectable break in the slope of $\mathcal{R}_t$ in the three-week period after the introduction of the lockdown (i.e., no "kink"). In S1 Appendix (Section A.13, Fig A.10 to Fig A.13), we show that other NPIs follow a similar pattern. In particular, we document the behavior of $\mathcal{R}_t$ around the introduction of public-event bans, case-based measures (such as self-isolation whenever feeling ill and experiencing fever), school closures, and social-distancing measures. Except for school closures and public-event bans, there is no visually apparent break in the trend of $\mathcal{R}_t$ around the date of the policy intervention.

To further investigate the behavior of $\mathcal{R}_t$ in the four-week window around lockdowns, we use mobility data from Google's "COVID-19 Community Mobility Reports" [24]. Google uses smartphone location data to measure changes in mobility (relative to pre-pandemic levels) for six six types of places: (i) groceries and pharmacies; (ii) parks; (iii) transit stations; (iv) retail and recreation; (v) residential; and (vi) workplaces. Since these measures are strongly correlated, we take the first principal component of the six time series to construct an overall mobility index. The first principal component explains 83.03% of the total variation in Google's mobility data.

Fig 4 shows that most of the decline in mobility occurs *before* the imposition of the lockdown, and remains low *thereafter*. This finding shows a clear change in people's behavior in

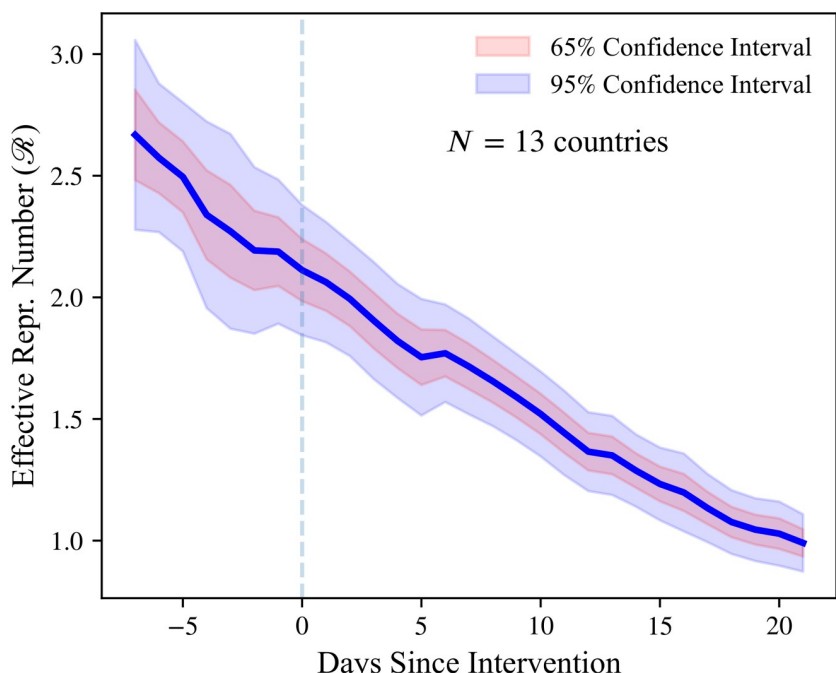

**Fig 3. $\mathcal{R}$ and policy interventions: Lockdowns.** Estimated effective reproduction number ($\mathcal{R}_t$) one week before and three weeks after a lockdown is introduced in a country. The original sample consists of 14 European countries studied by Flaxman et al [25]. Heteroskedasticity-robust confidence bounds are shown by the shaded areas.

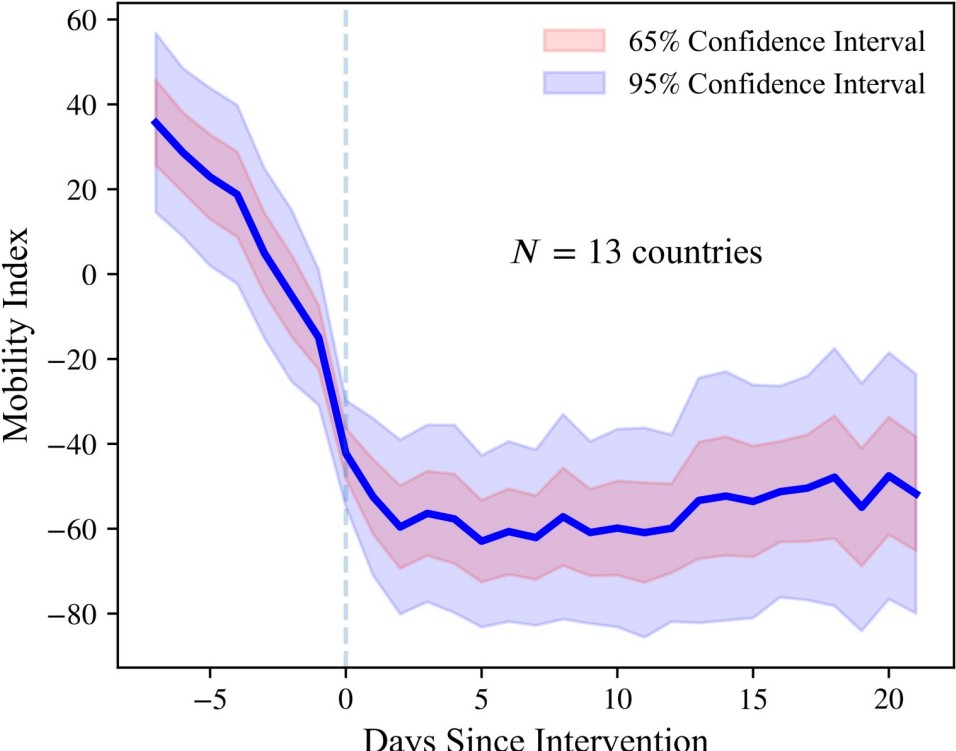

**Fig 4. Mobility around introduction of lockdowns.** Mobility index (constructed from Google's "COVID-19 Community Mobility Reports" [24]) one week before and three weeks after a lockdown is introduced in a country. See S1 Appendix (Section A.8) for details on the construction of the mobility index. The original sample consists of 14 European countries studied by Flaxman et al [25]. Heteroskedasticity-robust confidence bounds are shown by the shaded areas.

the early days of the pandemic. Shifting habits before the introduction of NPIs is consistent with the existence of private motives that can induce a reduction in mobility as people avoid becoming infected [48–50]. Our results are also consistent with empirical evidence for the U.S and anecdotal reports from Sweden [48, 51]. The documented relationship between $\mathcal{R}_t$ and mobility does not necessarily constitute evidence against the effectiveness of lockdowns. On the contrary, it is possible that lockdowns reinforce attitudes towards disease-awareness and self-isolation, helping to ensure lower values of $\mathcal{R}_t$ in the long run.

A potential concern with the evidence in Fig 3 is that our estimates of $\mathcal{R}_t$ use information from the full sample. Hence, estimates of $\mathcal{R}_t$ *before* the lockdown implicitly depend on the estimates of $\mathcal{R}_t$ *after* the lockdown. This feature of the estimation procedure may result in low statistical power to detect any effects of NPIs. To investigate this possibility, we conduct a power analysis (see Section A.11 in S1 Appendix). Given our empirical estimates of signal-to-noise ratios, we find that the statistical procedure appears sufficiently powerful to detect moderate changes in $\mathcal{R}_t$.

To assess the effects of NPIs more formally, we employ the following fixed-effect regressions (Table 2). Specifically, we regress $\mathcal{R}_t$ on a set of indicator variables capturing interventions and different types of fixed effects:

$$\log\left(R_{i,t}\right) = (\text{fixed effects}) + \sum_{j=1}^{5} \beta_i \text{NPI}_{i,t}^{(j)} + u_{i,t},$$

**Table 2. Effective reproduction number after introduction of NPIs.**

| | Dependent Variable: $log(R_t)$ | | | |
| | (1) | (2) | (3) | (4) |
|---|---|---|---|---|
| Lockdown | -0.65*** | -0.13** | -0.1** | -0.04 |
| | (0.04) | (0.05) | (0.05) | (0.06) |
| Public Events | -0.09* | 0.22*** | 0.21*** | 0.36*** |
| | (0.04) | (0.04) | (0.04) | (0.08) |
| School Closure | -0.02 | -0.19*** | -0.18*** | -0.1 |
| | (0.06) | (0.05) | (0.06) | (0.08) |
| Self Isolation | -0.13** | -0.09** | -0.04 | 0.01 |
| | (0.06) | (0.05) | (0.05) | (0.08) |
| Social Distancing | -0.15** | -0.08 | -0.11* | -0.14* |
| | (0.06) | (0.06) | (0.06) | (0.08) |
| $N$ | 855 | 855 | 848 | 488 |
| $R^2$ | 0.53 | 0.87 | 0.88 | 0.91 |
| Country FE | ✓ | ✓ | ✓ | ✓ |
| Days-Since-Outbreak FE | | ✓ | ✓ | ✓ |
| Mobility Controls | | | ✓ | ✓ |
| Testing Controls | | | | ✓ |

* $p < 0.1$;

** $p < 0.05$;

*** $p < 0.01$

Results of panel-data regressions of the (log of) effective reproduction number ($\mathcal{R}_t$) on indicator variables that are equal to 1 after the introduction of a non-pharmaceutical intervention (NPI) and 0 before the introduction. The sample consists of 14 European countries studied by Flaxman et al [25]. Regressions always include country fixed effects; regressions in columns (2)–(4) also include days-since-outbreak fixed effects. Outbreak is defined as the date on which 100 cases of COVID-19 are reached. The regression with mobility controls in (3) includes the one- and two-week lags of the mobility index; see S1 Appendix (Section A.8) for details. The regression with testing controls in (4) controls for the change in the number of daily tests per capita conducted in the country. To allow for reasonably precise estimation of days-since-outbreak fixed effects, we only consider days after the outbreak for which we have data for at least 5 countries. Heteroskedasticity-robust standard errors in parentheses.

where $u_{i,t}$ denotes the stochastic error term of the regression. The $\mathrm{NPI}_{i,t}^{(j)}$ is an indicator variable that equals 1 after the $j$-th NPI is introduced, and zero before its introduction. The index $i$ denotes countries, and $t$ stands for the number of days since the outbreak of the epidemic.

Column (1) of Table 2 provides estimated effects of NPIs when only country fixed effects are included. We observe a strong negative effect of lockdowns, social distancing, and measures of self isolation. Taken at face value, the estimates suggest that lockdowns reduce $\mathcal{R}_t$ by 65%. School closures are not statistically significant in this specification. These regressions as well the point estimates are similar to the statistical analysis performed by Flaxman et al [25].

The regression with country fixed effects only, however, is likely misspecified. Implicitly, such a specification assumes that the only reason $\mathcal{R}_t$ can fall is because of introduction of NPIs. However, $\mathcal{R}_t$ would likely trend downwards even in the absence of any public policy interventions. First, $\mathcal{R}_t$ tends to fall during an epidemic as the number of susceptibles is depleted. Second, people may adjust their behavior even in the absence of any policy measures. Failing to control for the dynamics of $\mathcal{R}_t$ in the absence of NPIs therefore likely leads to an over-estimation of the effects of NPIs.

We acknowledge that obtaining credible counterfactuals in the present empirical context is extremely challenging. However, we can exploit the panel structure of the dataset to reduce the potential issues in the previous specification. We do so by including days-since-outbreak fixed

effects. Intuitively, with such fixed effects we are comparing $\mathcal{R}_t$'s in two countries (e.g., country A and country B) that are both five days from the outbreak (say), with a school closure in country A but not in country B.

The results from the regression with days-since-outbreak fixed effects are shown in column (2). The coefficient for lockdowns becomes substantially smaller in absolute value and less statistically significant. The coefficients for self-isolation and social-distancing measures are also reduced and lose some of their statistical significance. The coefficient for public events is highly statistically significant but positive rather than negative. A naïve interpretation would suggest that banning public events has a positive effect on $\mathcal{R}_t$. More likely, however, is that the positive coefficient is due to countries where $\mathcal{R}_t$ is declining more slowly being faster to ban public events. In S1 Appendix (Section A.14), we show that the results remain similar when the NPIs are included separately, reducing concerns about potential multicollinearity problems between the different NPI variables.

In column (3), we also include lagged mobility variables as additional controls. With mobility controls, the coefficient on lockdowns is further reduced. School closures and social-distancing measures are estimated to have a statistically-significant negative effect on $\mathcal{R}_t$, with reducing $\mathcal{R}_t$ by 18% and 11%, respectively.

A potential concern is that countries may introduce NPIs and simultaneously increase the number of tests for COVID-19 that they perform. To help alleviate this concern, in column (4) we add the change in the daily number of tests per capita as an additional explanatory variable. The data on daily tests per capita comes from *Our World in Data* [23]. With testing controls, most coefficients are no longer statistically significant. Note, however, that the sample size is reduced significantly as we do not have testing data for all countries in the sample.

We caution readers against over-interpreting the results of this section. Obtaining unbiased estimates of the true causal impact of NPIs is exceptionally challenging. As a result, even our best estimates might still suffer from statistical issues such as unobservable confounding variables or simultaneity bias. In particular, the timing of NPIs is not random. Countries that introduced NPIs earlier likely did so because they had previously observed a stubbornly high $\mathcal{R}_t$. In that case, the dependent and independent variables would be simultaneously determined, yielding biased estimates. Moreover, since we cannot directly observe peoples' attitudes towards COVID-19 or government policies, we cannot control for other variables affecting human behavior. These potential issues notwithstanding, we find that people adjusted their mobility patterns *before* the introduction of lockdowns. We believe that these findings bolster the importance of accounting for changes in human behavior when evaluating the effects of NPIs.

## Conclusion

In this paper we develop a new way to estimate the effective reproduction number of an infectious disease ($\mathcal{R}$). Our estimation method exploits a structural mapping between $\mathcal{R}$ and the growth rate of the number of infected individuals derived from the basic SIR model. The new methodology is straightforward to apply in practice, and according to our simulation checks, it yields accurate estimates. We use the new method to track $\mathcal{R}$ of COVID-19 around the world, and assess the effectiveness of public policy interventions in a sample of European countries.

The current paper faces several limitations. First, a local-level specification for the growth rate implicitly assumes that the growth rate of the number of infected individuals remains forever in flux. However, in the long-run, this growth rate must converge to zero. Since our model does not capture this feature, it seems likely that our estimated confidence bounds are overly conservative in the late stages of an epidemic. Second, when applying the model to

cross-country data, one may achieve important gains in statistical efficiency if the model is estimated jointly for all countries (for example, by estimating a multivariate local-level model). Finally, for assessing the effects of NPIs more accurately, it would be desirable to collect data for a larger sample of countries.

Our estimates of $\mathcal{R}$ for COVID-19 are based on a structural relationship derived from the SIR model. By using the SIR model, we omit some features of the disease that are likely important when modeling its spread. In particular, the SIR model abstracts away from incubation periods as well as transmission during the incubation period. Nevertheless, we prefer the SIR specification, for two reasons. First, in simulations, we find that our estimator produces accurate estimates even when the true model is SEIR rather than SIR, as we show in S1 Appendix (Section A.5). Second, we believe that the SIR model is likely to produce more reliable estimates in practice. To use the SEIR model, we would have to estimate the number of currently exposed individuals. Doing so would triple the number of model parameters. In particular, we would have to calibrate the (i) average duration of the incubation period ($\kappa^{-1}$); and (ii) relative infectiousness of exposed and infectious individuals ($\epsilon$); see S1 Appendix (Section A.5) for details. While $\kappa$ is arguably constant across countries, $\epsilon$ is unlikely to be fixed over countries and over time. For example, greater mask usage is likely to reduce $\epsilon$ by differentially affecting transmission by symptomatic and pre- or asymptomatic individuals. Allowing for such time variation in $\epsilon$, in addition to time-varying transmission rates ($\beta_t$), is challenging. That said, it is possible to extend this paper's ideas to models that are richer than the SIR model. Doing so may be an exciting avenue for future research.

Relative to existing methods for estimating $\mathcal{R}$, we combine basic epidemiological theory with standard time-series filtering techniques, particularly Kalman filtering. This approach leads to a transparent closed-form estimator. The simplicity of the estimator allows us to study some of its properties analytically (e.g., the effects of potential data problems). Differently from most existing approaches, our method can be applied using both Bayesian and frequentist techniques, and it does not require any tuning parameters beyond specifying the average serial interval. On the other hand, relative to less structural approaches such as that of Cori et al [3], our estimator may be more sensitive to potential model misspecification. Empirically, we find that our estimates and estimates obtained by the Cori et al method are highly positively correlated (average correlation: 0.80). However, the correlations are not perfect, suggesting that there is value in combining both estimators when tracking infectious diseases. Hence, our methodology brings an additional instrument to the researcher's toolbox.

In our empirical application, we find that lockdowns, measures of self-isolation, and social distancing all have statistically significant effects on reducing $\mathcal{R}$ of COVID-19. However, we also demonstrate the importance of accounting for voluntary changes in behavior. In particular, most of the decline in mobility in our sample took place before lockdowns were introduced. This finding suggests that people respond to the risk of contracting the virus by changing their mobility patterns and reducing social interactions. Failing to account for such voluntary changes in behavior yields estimated effects of NPIs that are arguably too large.

Given that even our best estimates may still be biased, it is important to interpret these results cautiously. However, from an economic perspective, these findings point to large private incentives to avoid infection. These incentives can induce a contraction in economic activity as people voluntarily choose to self-isolate [48–50]. As a result, even if countries lift the NPIs that are currently in place, it is not clear whether people would voluntarily return to their pre-pandemic mobility and consumption patterns. Our real-time estimator may be used to track the dynamics of COVID-19 as the current restrictions are relaxed.

## Supporting information

**S1 Appendix. Supplementary appendix.** Supplementary appendix containing additional theoretical and simulation results, data descriptions, and further empirical analysis.
(PDF)

**S1 File. Replication files.** Replication package for replicating all of the statistical analysis and simulation results provided in the paper.
(ZIP)

## Acknowledgments

We would like to thank Christiane Baumeister, Ralph Brinks, Eric Budish, Ricardo Hausmann, Artūras Juodis, Jan Keil, Siem Jan Koopman, Andrés Neumeyer, Mathieu Pedemonte, Sergio Ocampo-Díaz, Sándor Sóvágó, Eduardo Undurraga, Iván Werning, as well as seminar participants at the Central Bank of Chile and Harvard Growth Lab for their comments and suggestions. We also thank Chen Lin for excellent research assistance. The views and conclusions presented in this paper are exclusively those of the authors and do not necessarily reflect the position of the Central Bank of Chile or of the Board members.

## Author Contributions

**Investigation:** Francisco Arroyo-Marioli, Francisco Bullano, Simas Kucinskas, Carlos Rondón-Moreno.

**Methodology:** Francisco Arroyo-Marioli, Francisco Bullano, Simas Kucinskas, Carlos Rondón-Moreno.

**Software:** Francisco Arroyo-Marioli, Francisco Bullano, Simas Kucinskas, Carlos Rondón-Moreno.

**Writing – original draft:** Francisco Arroyo-Marioli, Francisco Bullano, Simas Kucinskas, Carlos Rondón-Moreno.

**Writing – review & editing:** Francisco Arroyo-Marioli, Francisco Bullano, Simas Kucinskas, Carlos Rondón-Moreno.

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
