## [Decision Letter · Decision Letter 0]

18 Aug 2020

PONE-D-20-13789

Tracking R of COVID-19: a new real-time estimation using the Kalman filter

PLOS ONE

Dear Dr. Kucinskas,

Thank you for submitting your manuscript to PLOS ONE. After careful consideration, we feel that it has merit but does not fully meet PLOS ONE’s publication criteria as it currently stands. Therefore, we invite you to submit a revised version of the manuscript that addresses the points raised during the review process.

We look forward to receiving your revised manuscript.

Kind regards,

Benn Sartorius, PhD

Academic Editor

PLOS ONE

Additional Editor Comments:

The paper proposes a new method for estimating the effective reproduction number of an infectious disease and apply it to track the dynamics of COVID-19 using data from 124 countries. Furthermore they have also used this framework to assess the effectiveness of non-pharmaceutical

interventions in a sample of 14 European countries. The analysis appears to be very rigorous and the paper well written.

Comments:

-Please include a completed GATHER checklist (http://gather-statement.org/) and include in your supplementary material as well as make reference to this at the beginning of your methods section.

-My understanding for COVID-19, is that a SEIR model should be used instead of the SIR. The authors refer to the SEIR formulation in the supplementary as part of their sensitivity analyses but I wonder why the main results were not just based on the SEIR assuming duration of infectiousness based on a published estimates/uncertainty range?

-Discussion: please include more comparison/contrast with other similar published reproductive number estimates available in the literature.

'The author(s) received no specific funding for this work.'

We note that one or more of the authors are employed by a commercial company: Central Bank of Chile.

Reviewers' comments:

Reviewer's Responses to Questions

**Comments to the Author**

1. Is the manuscript technically sound, and do the data support the conclusions?

Reviewer #1: Partly

2. Has the statistical analysis been performed appropriately and rigorously? 

Reviewer #1: Yes

3. Have the authors made all data underlying the findings in their manuscript fully available?

Reviewer #1: Yes

4. Is the manuscript presented in an intelligible fashion and written in standard English?

Reviewer #1: Yes

5. Review Comments to the Author

Reviewer #1: Comments to the authors

This manuscript is a useful addition to primary research on estimating the effective reproduction number of infectious diseases, applied to the ongoing COVID-19 pandemic. The method presented is said to produce estimates of both the instantaneous and cohort/case reproduction number which is advantageous. The conclusions drawn from applying the method to COVID-19 are in the most part sound and well explained though some important minor changes and clarifications around the effect of lockdown are advised below.

Detailed feedback:

Abstract

“the method is very easy to apply in practice” –consider being more specific about what makes it easy to apply

Main text

Repeated throughout - when referencing material in the SI, the authors should include the section number and figure number to help the reader quickly locate the material.

Line 2. At the beginning the authors should stress the difference between R0 and effective reproduction number and include references for standard definitions. This difference is highlighted in Lines 82 through 84. However, it would be better to move this up in the introduction to give a better background.

Lines 36 to 39. References that support each of the approaches outlined should be cited here.

Line 60. It would be good to be more specific/formal about what is meant by “well-behaved estimates”

Lines 112. "As is well known, the local-level model is sufficiently flexible to capture rich dynamic patterns in the data." The authors should give some references here to support this assertion to help readers.

Line 160. A sentence about the sensitivity of your results to your chosen serial would be good here to summarise what you show in the supplementary (particularly as you note show in the SI that having an accurate estimate for the this parameter is required for your estimator to be robust to model misspecifications)

Line 168 & Figure 1. Is it meaningful to present Rt for the whole world aggregated together? Might be better to present more of your country level estimates.

Figure 2. Since different colours are used to denote Rt estimates for China, Italy and the US, there is no need to also use different linetypes.

Line 181. Add reference to the Figure number in the SI.

Line 187. "..but at a slower rate than previously observed in China". Can the authors quantify this rate? For example, the time difference between the start of the epidemic (as defined in this paper) and the first time point estimate of R is below 1?

Line 195. The statement will be strengthened if the authors provide the estimate with confidence bounds. A similar comment applies when qualitative statements are made e.g., "somewhat", "almost" etc.

Line 208. Add reference to the exact section number and figure that the reader should check.

Line 236. "Event study analysis" I do not think that this is a commonly used terminology in epidemiology, and the authors should provide a brief definition to help the reader.

Line 238. Present 95% confidence intervals alongside these point estimates.

Line 239. "There is no visually detectible break in the slope of R around the introduction of lockdown". Can the authors quantify what they mean by "around the time of lockdown"? Any change in policy or behaviour is not expected to have an immediate effect on R, and this should be taken into consideration. This comment also applies to Line 247 where the authors state that "R_t appears to be unaffected by lockdown." Consider rephrasing this. No counterfactual is available, and just because behaviour changed before lockdown doesn’t mean it would have remained changed in the absence of a lockdown as you later mention. In addition, peoples initial behavioural change may have been influenced by discourse around potential upcoming lockdown and lockdowns introduced in other countries prior to introduction in their own country.

Line 247. A brief description of the construction of the mobility index should be included in the main text. Can the authors check also that the decline in mobility before lockdown is consistent across all countries considered, and for other combinations e.g. countries in Asia vs countries in Europe? Finally, my comment about qualitative statements is applicable to the description in SI ("the first principal component explains a little less than 85% of the total variance in the data"). Perhaps state the actual variance explained?

Line 266. State what is u_{i, t}.

Line 276. The only reason Rt can fall is because of... omit "why"

Table 2. If you could add concise titles to the columns in place of (1) to (4) it would be more immediately clear to the reader.

Line 312. Give a brief definition/detail on what is meant by endogeneity problems or use a different phrase – it is not guaranteed to be terminology that readers will have come across, though they will be familiar with the problem itself.

Supplementary material

Page 1 sentence one “also obtains” should be “is also obtained”

SI Section 6 page 8. This section is very interesting – inconsistent levels of reporting is a major issue in Rt estimation. I would make sure readers are clearly signposted to this section from the main text when you talk about data problems there.

SI page 18. "For both validation exercises performed in the present section, we include all countries for which we have at least 20 observations after the onset of the epidemic (100 cumulative cases of COVID-19 reached)." Can the authors state how many countries were included? Figure 4 states N =13, but I believe this means only the European countries considered minus one?

6. PLOS authors have the option to publish the peer review history of their article (what does this mean?). If published, this will include your full peer review and any attached files.

Reviewer #1: No

---

## [Author Response · Author response to Decision Letter 0]

16 Nov 2020

Dear Editor,

We would like to sincerely thank you for the opportunity to revise our manuscript entitled “Tracking R of COVID-19: A New Real-Time Estimation Using the Kalman Filter.” In addition, we are grateful to the Referee for their insightful and constructive suggestions on how to improve the paper.

All comments raised by you and the Referee have been addressed in the revised manuscript. Most suggestions have been incorporated in the main body, while a few are addressed in the Supplementary Appendix to avoid overextending the main paper.

To avoid any possible misunderstanding, we would like to clarify that the Central Bank of Chile is a public institution owned and supervised by the Chilean government.

Point-by-point responses are provided in the attached files.

Dr. Francisco Arroyo-Marioli

Francisco Bullano,

Dr. Simas Kučinskas,

Dr. Carlos Rondón-Moreno

---

## [Editor Report · Decision Letter 1]

11 Dec 2020

Tracking R of COVID-19: a new real-time estimation using the Kalman filter

PONE-D-20-13789R1

Dear Dr. Kucinskas,

We’re pleased to inform you that your manuscript has been judged scientifically suitable for publication and will be formally accepted for publication once it meets all outstanding technical requirements.

Kind regards,

Benn Sartorius, PhD

Academic Editor

PLOS ONE
---

## [Editor Report · Acceptance letter]

22 Dec 2020

PONE-D-20-13789R1

Tracking *R* of COVID-19: a new real-time estimation using the Kalman filter

Dear Dr. Kucinskas:

I'm pleased to inform you that your manuscript has been deemed suitable for publication in PLOS ONE. Congratulations! Your manuscript is now with our production department.

Kind regards,

on behalf of

Dr. Benn Sartorius

Academic Editor

PLOS ONE